# Sunitinib in Patients with Metastatic Renal Cell Carcinoma with Favorable Risk: Be Aware of PD-L1 Expression

**DOI:** 10.3390/medsci12030048

**Published:** 2024-09-13

**Authors:** Ilya Tsimafeyeu

**Affiliations:** Bureau for Cancer Research—BUCARE, 526 W 158th Str., New York, NY 10032, USA; tsimafeyeu@gmail.com

**Keywords:** metastic renal cell carcinoma, first-line therapy, sunitinib, immunotargeted therapy, PD-L1 expression, favorable IMDC risk

## Abstract

The treatment landscape for metastatic renal cell carcinoma (RCC) has advanced significantly with first-line immunotargeted therapy combinations. However, no statistically significant differences were observed in the cohort of patients with favorable risk and some oncologists continue to use sunitinib in these patients. PD-L1 expression has emerged as a negative prognostic factor in RCC, particularly in sunitinib-treated patients, where higher PD-L1 levels are linked to worse outcomes. This article discusses the potential risks associated with the use of sunitinib in PD-L1-positive patients.

Progress in the treatment of metastatic renal cell carcinoma (RCC) is evident: first-line dual immunotherapy and immunotargeted therapy have made significant contributions. Effective subsequent lines address the initial shortcomings, making the therapies more universal and suitable for any patient. Five combinations have been approved by regulatory agencies in various countries based on randomized phase 3 trials comparing their efficacy to sunitinib (CheckMate 214 [1], KEYNOTE-426 [2], Javelin Renal 101 [3], CheckMate 9ER [4], and CLEAR [5]). In their latest article, R. Motzer et al. published their final overall survival (OS) results for patients with metastatic RCC treated with first-line lenvatinib and pembrolizumab in the CLEAR study [5]. Overall, the OS results were significantly better in the immunotargeted therapy group compared to sunitinib (HR = 0.79). However, no statistically significant differences were observed in the cohort of patients with favorable risk (HR = 0.94). Similar findings have been reported with other combinations, leading many experts to believe that sunitinib can be used as a first-line therapy option in patients with favorable or very favorable risk, with its disadvantages mitigated by subsequent lines of therapy, thereby not worsening survival [6]. Given that the studies were designed to demonstrate differences in the OS or even progression-free survival (PFS) in the intent-to-treat population, this idea may be controversial. It is unknown how the risk groups, including those with favorable risk, were balanced for other factors such as PD-L1 expression. Since PD-L1 positivity was used in some of the trials, this heterogeneity could have influenced the results. PD-L1 expression on tumor tissue has been reported in 25–60% of patients, depending on the assay used [7]. Generally, PD-L1 expression has been identified as a negative prognostic factor in metastatic RCC [8]. Additionally, high PD-L1 levels are associated with unfavorable outcomes for tyrosine kinase inhibitors [9]. Therefore, combining an anti-PD-1/PD-L1 antibody with targeted therapy leverages several potentially synergistic mechanisms.

Among these trials, only one positive study, Javelin Renal 101, was specifically designed to demonstrate the effectiveness of immunotargeted therapy in a population of patients expressing PD-L1. However, all other studies included between 22% and 63% of patients with immunohistochemical PD-L1 expression, assessed using different antibody clones (DAKO 28-8 [10,11], DAKO 22C3 [12,13], and Ventana SP263 [14]) and various models (expression on tumor and/or immune cells). The study and control arms were well balanced for PD-L1 expression in the ITT population. Fortunately, subgroup analyses showed an approximately equal efficacy of the new combinations in both PD-L1-positive and PD-L1-negative patients, indicating that verifying PD-L1 status before initiating immunotherapy is unnecessary [10,11,12,13,14].

However, this is not the case for sunitinib. Previously, in the randomized phase 3 COMPARTZ trial, the authors demonstrated a statistically significant association between PD-L1 expression and worse OS outcomes in first-line sunitinib therapy. The median OS was 15.3 months in PD-L1-positive and 27.8 months in PD-L1-negative cohorts [15]. In a multivariate analysis, PD-L1 expression was identified as an independent prognostic indicator of poor OS.

The same trend was observed in immunotherapy trials where sunitinib was studied as a control (Table 1). In the CheckMate 214 trial, the median PFS and OS were two-fold higher in PD-L1-negative patients compared to PD-L1-positive patients treated with sunitinib [10]. Specifically, the median OS was 42.22 and 23.88 months, respectively. Similarly, in the KEYNOTE 426 study, patients with PD-L1 expression had a significantly shorter OS in the sunitinib group (*p* = 0.025) [13]. Negative associations between PD-L1 expression and PFS were also observed in the Javelin Renal 101 (*p* = 0.0037) and CheckMate 9ER (*p* = 0.00045) trials [11,14]. In the CLEAR study, patients with CPS < 1 showed a trend towards a longer PFS in the sunitinib arm (*p* = 0.067) [12]. Biomolecular analysis revealed that PD-L1-positive tumors were enriched in immune/proliferative, proliferative, and stromal/proliferative clusters. These mixed signatures were associated with a worse outcome with sunitinib and a worse prognosis. Finally, in the IMmotion151 study, despite its negative outcome, it was observed that patients in the sunitinib group with PD-L1 expression had a reduced PFS, particularly when PD-L1 expression was associated with the sarcomatoid features, with a median of 8.4 months in the ITT population compared to 5.6 months in the cohort with PD-L1-positive sarcomatoid RCC [16].

As a conclusion, it should be noted that multiple studies have shown a reduced efficacy of sunitinib in patients with PD-L1 expression. This suggests that PD-L1 expression could be a negative predictive factor for sunitinib treatment, aligning with broader findings that PD-L1 positivity is associated with poorer outcomes in metastatic RCC. Secondly, there are limited data on the impact of PD-L1 status across different IMDC risk groups. Despite the recognition of PD-L1 as a negative biomarker, there remains limited evidence on how PD-L1 status influences the activity of sunitinib in patients with favorable risk. Although there is currently no strong evidence supporting the use of PD-L1 alone as a biomarker for sunitinib treatment in metastatic RCC in daily practice, it is crucial to exercise caution in asserting that sunitinib can be safely administered in groups with favorable risk. These patients may still express PD-L1, potentially compromising the effectiveness of targeted therapy. This complexity underscores the need for more nuanced and personalized approaches to RCC treatment, incorporating a broader range of biomarkers and patient characteristics to guide therapy decisions more effectively. Moreover, future research should focus on elucidating the role of PD-L1 and other potential biomarkers in predicting responses to targeted and immunotherapies across diverse patient populations. This would enhance our ability to tailor treatments to individual patients, maximizing efficacy and improving the overall outcomes in metastatic RCC.

## Figures and Tables

**Table 1 medsci-12-00048-t001:** Efficacy of sunitinib depending on PD-L1 expression in randomized trials of first-line therapy for metastatic clear cell RCC.

Study	PFS Median, Months			OS Median, Months		
	PD-L1 Negative	PD-L1 Positive	*p*	PD-L1 Negative	PD-L1 Positive	*p*
CheckMate 214 [10]	11.27	5.72	NR	42.22	23.85	NR
KEYNOTE 426 [13]	NR	NR	NS	NR	NR	0.025 (PD-L1 expression was negatively associated with the OS)
Javelin Renal 101 [14]	11.1	8.2	0.0037	NR	36.2	NR
CheckMate 9ER [11]	NR	NR	0.00045 (PD-L1 expression was negatively associated with the PFS)	NR	NR	NR
CLEAR [12]	NR	NR	0.067	NR	NR	NR
IMmotion151 [16,17]	8.4	5.6	NR	NR	31.6	NR

PFS, progression-free survival; OS, overall survival; NR, not reported; NS, non-significant; and *p*, *p*-value.

## Data Availability

No new data were created or analyzed in this study.

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
