# Peer review of "Sunitinib in Patients with Metastatic Renal Cell Carcinoma with Favorable Risk: Be Aware of PD-L1 Expression"

_medsci, 2024, doi:10.3390/medsci12030048_

Round 1

Reviewer 1 Report

Comments and Suggestions for Authors

Tsimafeyeu have produced a short commentary regarding TKI treatment in good prognosis patients related to PD L1 status.

This short report provides a valuable and important contribution in a time when focus clearly lies on different lines of IO.

The report meets all criteria to be published. Minor (very minor) comments below.

Statement on line 28-29 should probably be referenced

Remove "of" on line 35

Author Response

Dear colleagues,

We thank our reviewers for detailed review of our manuscript and critical comments.  We have carefully addressed every comment.  We truly believe this revision has strengthened our manuscript.

Below is the line-by-line response to the reviewers’ comments.

1) Statement on line 28-29 should probably be referenced

Reply: I fully agree with the reviewer. I have added this reference. All references were re-numbered.

2) Remove "of" on line 35

Reply: We are extremely grateful to the reviewer for recommendation. I removed it.

Reviewer 2 Report

Comments and Suggestions for Authors

The topic is about the connection of Sunitinib in Metastatic Renal Cell Carcinoma Patients with favorable Risk: Be Aware of PD-L1 Expression.

The commentary is very interesting and informative for urologist in the field.

I have only two question:

1. What about the PD-L1 expression in primary renal cancer and how the serum PD-L1 level correlates with the tissues PD-L1 amount?

2. Would other type of tyrosin-kinase drug based treatment be also effected by the PD-L1 level in patient diagnosed with renal cancer?

Author Response

We thank reviewer for detailed review of our manuscript and critical comments.  We have carefully addressed every comment. Below is the line-by-line response to the reviewer’ comments.

What about the PD-L1 expression in primary renal cancer and how the serum PD-L1 level correlates with the tissues PD-L1 amount?

Reply: We thank reviewer for these insightful comments. We have clarified in text: PD-L1 expression on tumor tissue has been reported in 25–60% of patients. Unfortunately, there were no clinical studies for novel immunotargeted combinations including correlations between tissue and plasma PD-L1 level.

Would other type of tyrosin-kinase drug based treatment be also effected by the PD-L1 level in patient diagnosed with renal cancer?

Reply: Thanks for the important question. Other tyrosine kinase inhibitors were not comparators in the studies, so unfortunately no information is available. In our article, we focused on sunitinib as the only comparator in the phase 3 trials.

Reviewer 3 Report

Comments and Suggestions for Authors

This is a well-written short communication describing the well-known, yet relatively  forgotten inverse relationship between PDL1 expression and survival  in mRCC patients under sunitinib treatment. 

The results presented support the  argument about the importance of PDL1 in this setting.

However, the results of the IMmotion 151 trial, which enrolled patients depending on their PDL1 status, are not mentioned, nor is the possible implication of PDL1 positivity across the molecular subtypes defined by the analysis of this trial. Given the relevance of this commentary on the PDL1 status and the design of this trial, I encourage a report of this trial as well.

Author Response

Many thanks to the reviewer for the important comment! Indeed, the IMmotion151 study had a large group of patients with PD-L1 expression who received sunitinib. I have added this information to the text and table.

Reviewer 4 Report

Comments and Suggestions for Authors

The commentary titled ‘Sunitinib in Metastatic Renal Cell Carcinoma Patients with Favorable Risk: Be Aware of PD-L1 Expression’ by Ilya Tsimafeyeu is clinical in nature, timely, and highly relevant.

Sunitinib, a small molecule kinase inhibitor targeting the Vascular Endothelial Growth Factor receptor (VEGFR), is a key treatment for advanced-stage renal cell carcinoma (RCC). In this commentary, the author discusses five important clinical trials (CheckMate, KEYNOTE, Javelin Renal, CheckMate 9ER, and CLEAR) that have been approved by regulatory agencies in various countries based on randomized phase 3 trials comparing their efficacy with sunitinib.

The author comments on PD-L1 expression and its association with clinical outcomes reported in these trials and other studies. Additionally, the commentary highlights the correlation between PD-L1 expression and the efficacy of sunitinib in treating RCC.

Overall, the commentary is insightful and may be considered for publication in its current form.

Author Response

I am very grateful for the reviewer's careful reading of our article and assessment!